# Molecular Cloning, Expression Analysis, and Functional Analysis of Nine *IbSWEETs* in *Ipomoea batatas* (L.) Lam

**DOI:** 10.3390/ijms242316615

**Published:** 2023-11-22

**Authors:** Jingli Huang, Xuezhen Fu, Wenyan Li, Zhongwang Ni, Yanwen Zhao, Pinggang Zhang, Aiqin Wang, Dong Xiao, Jie Zhan, Longfei He

**Affiliations:** 1College of Agriculture, Guangxi University, Nanning 530004, China; hjl19871011@163.com (J.H.); fxz1773319769@163.com (X.F.); liwenyan20100@163.com (W.L.); nzw20011106@163.com (Z.N.); 15678585108@163.com (Y.Z.); 13006915700@163.com (P.Z.); waiqing1966@126.com (A.W.); xiaodong@gxu.edu.cn (D.X.); may2399@163.com (J.Z.); 2Agricultural and Animal Husbandry Industry Development Research Institute, Guangxi University, Nanning 530004, China; 3Guangxi Key Laboratory of Agro-Environment and Agro-Product Safety, College of Agriculture, Guangxi University, Nanning 530004, China; 4Key Laboratory of Crop Cultivation and Tillage, College of Agriculture, Guangxi University, Nanning 530004, China

**Keywords:** *Ipomoea batatas*, SWEET, expression analysis, protein interaction, abiotic stress

## Abstract

Sugar Will Eventually be Exported Transporter (*SWEET*) genes play an important regulatory role in plants’ growth and development, stress response, and sugar metabolism, but there are few reports on the role of SWEET proteins in sweet potato. In this study, nine *IbSWEET* genes were obtained via PCR amplification from the cDNA of sweet potato. Phylogenetic analysis showed that nine IbSWEETs separately belong to four clades (Clade I~IV) and contain two MtN3/saliva domains or PQ-loop superfamily and six~seven transmembrane domains. Protein interaction prediction showed that seven SWEETs interact with other proteins, and SWEETs interact with each other (SWEET1 and SWEET12; SWEET2 and SWEET17) to form heterodimers. qRT-PCR analysis showed that *IbSWEETs* were tissue-specific, and *IbSWEET1b* was highly expressed during root growth and development. In addition to high expression in leaves, *IbSWEET15* was also highly expressed during root expansion, and *IbSWEET7*, *10a*, *10b*, and *12* showed higher expression in the leaves. The expression of *SWEETs* showed a significant positive/negative correlation with the content of soluble sugar and starch in storage roots. Under abiotic stress treatment, *IbSWEET7* showed a strong response to PEG treatment, while *IbSWEET10a*, *10b*, and *12* responded significantly to 4 °C treatment and, also, at 1 h after ABA, to NaCl treatment. A yeast mutant complementation assay showed that IbSWEET7 had fructose, mannose, and glucose transport activity; IbSWEET15 had glucose transport activity and weaker sucrose transport activity; and all nine IbSWEETs could transport 2-deoxyglucose. These results provide a basis for further elucidating the functions of *SWEET* genes and promoting molecular breeding in sweet potato.

## 1. Introduction

In plants, photosynthesis provides a material basis for plant growth and development, with up to 50–80% of photosynthates being transported from the phloem of the leaf to other organs to meet the needs of non-photosynthetic organs [1,2]. Sucrose is the main carbohydrate delivered by the source tissue to the sink tissue. Sugar transport is accomplished by sugar transporters, including monosaccharide transporters (MSTs) [3]; sucrose transporters (SUTs) [4,5]; and sugars-will-eventually-be-exported transporters (SWEETs) [6]. There are two different ways that sucrose enters the sink organ, the symplast pathway and the apoplast pathway [7,8]. The symplast pathway is accomplished via plasmodesmata, while the apoplast pathway occurs via two plasma-membrane-localized sugar transporters, SUTs and SWEETs [9]. In the apoplast pathway, sucrose is hydrolyzed into hexose (glucose and fructose), followed by hexose, using MSTs and SWEETs to complete source–sink transport [10]. 

SWEETs play important physiological roles in plants’ growth and development, stress response, and sugar metabolism [11]. *AtSWEET11* and *AtSWEET12* are important sucrose efflux transporters that co-operate to exert different physiological roles. Both transporters are involved in apoplast phloem loading, seed filling, and altered sugar levels at the site of pathogen infection [12,13]. Similarly, *SvSWEET4a* is a high-capacity glucose and sucrose transporter and is highly expressed in various maternal and offspring tissues within the seed, the vascular parenchyma of the peduncle and the xylem parenchyma of the stem, involved in the apoplast transport pathway of sink tissues [14]. *SWEET4* homologue [15,16,17], *SlSWEET12c* [18], *SlSWEET15* [19], *CsSWEET7a* [20], *XsSWEET10* [21], and others have also been reported to be involved in phloem unloading and post-phloem sugar transport pathways in sink tissues and usually have high expression levels in sink tissues. *NEC1* is a homologue of *SWEET9* and was found to play a key role in nectar secretion [22,23], with *AtSWEET13* and *AtSWEET14* mainly working at the anther wall later in development. The loss of *AtSWEET13* and *AtSWEET14* leads to reduced pollen viability, thus reducing pollen germination, but *SWEET9* completely rescued *atsweet13*, *14* pollen viability and germination defects; *AtSWEET13* and *AtSWEET14* may be responsible for sucrose efflux into the germ chamber, supporting pollen development to maturity [24]. In addition, *SlSWEET5b* [25], *JsSWEET9*, *JsSWEET2* [26], etc., are also associated with flower development. The plasma-membrane-localized MtN3 protein SAG29 (AtSWEET15) was mainly expressed in senescent tissues and was induced by osmotic stress through an abscisic-acid-dependent pathway and accelerated plant senescence [27]. Similarly, the overexpression of *OsSWEET5* and *PbSWEET4* led to accelerating senescence in rice and strawberry leaves, respectively, probably due to enhancing sugar export from the leaves [28,29]. *SWEETs* are also associated with plant grain filling and fruit enlargement [18,19,20,30,31], abiotic stress [32,33,34,35,36,37,38], and biological stress [39,40,41,42,43] in plants. 

Sweet potato is the seventh largest food crop in the world [44] and is an important cash crop. Its roots are rich in starch and soluble sugar, which can be used as food, feed, and industrial raw materials. Sweet potato is a highly heterozygous hexaploid plant, so the genetic study of sweet potato is very complicated. The development of genetics and genomics has enabled sweet potato and crop improvement to facilitate and accelerate the genetic improvement of this important rhizome crop through breeding, combining state-of-the-art multi-genomics such as genomic selection and gene editing. *Agrobacterium-tumefaciens*-mediated transformation of sweet potato provides the possibility of genome-modified breeding (including transgenic and genome editing breeding) [45]. Twenty-seven *IbSWEETs* were identified in sweet potato, which play multiple important roles in plant growth, storage root development, carotenoid accumulation, hormone crosstalk, and the abiotic stress response [46]. In addition, *IbSWEET10* could enhance the resistance of sweet potato to *Fusarium oxysporum* [47]; *IbSWEET15* increased the content of soluble sugars and starch in Arabidopsis seeds, while significantly reducing the soluble sugar content in leaves [48]. Considering the important role of *SWEETs* in plant development and to gain a more comprehensive and deeper understanding of the role of *SWEET* genes in the growth, development, and sugar accumulation of sweet potato, we conducted further research. In this study, nine *SWEETs* were screened for up/downregulation using RNA-seq datasets (submitted to NCBI with the accession number PRJNA678375), and we obtained the *SWEET* gene sequences from the sweet potato database (http://public-genomes-ngs.molgen.mpg.de/sweetpotato/ (accessed on 14 November 2023)). Nine *IbSWEETs* were obtained based on PCR amplification technology, and we performed gene characterization, protein structure, and transmembrane structure analysis. The qRT-PCR analysis indicated that they are tissue-specific and respond to abiotic stress. Heterologous expression in yeast revealed the substrates that transported by IbSWEET proteins. This study further clarified the function of IbSWEETs and promotes the molecular breeding of sweet potato.

## 2. Results

### 2.1. Molecular Cloning and Protein Characterization of IbSWEETs

To better understand the role of *IbSWEETs* in the growth and development process of sweet potato, we designed primers based on the *SWEETs* sequence obtained from the sweet potato genome and generated nine *IbSWEET* genes by PCR amplification from the cDNA of sweet potato (Appendix A). These *IbSWEET* genes were named *IbSWEET1a*, *1b*, *2*, *7*, *10a*, *10b*, *12*, *15*, and *17* according to the Arabidopsis homologous genes. The phylogenetic tree analysis indicated that nine IbSWEETs belong to four distinct clades (Clade I~IV) (Figure 1). These nine *IbSWEET* genes encode 234 to 305 amino acids; the theoretical isoelectric points (pI) ranged from 5.46 to 9.51; the molecular weight (MW) was between 25.61 to 34.36 (kD); the instability index (II) was between 24.29 and 51.82 and, among them, four IbSWEETs (IbSWEET1b, 7, 15, and 17) were less than 40, which were stable proteins, and the others were unstable proteins; the aliphatic index was 114.63–126.97; and all the hydrophobic indexes were greater than 0, so all IbSWEETs were hydrophobic proteins (Table 1). Subcellular localization prediction showed that IbSWEET1a and IbSWEET1b were localized on the plasma membrane, IbSWEET17 was localized on the vacuole membrane, and others were localized on chloroplasts (Table 1).

Conserved domain analysis showed that IbSWEETs contain two MtN3/saliva domains (CDD accession No. pfam03083) or PQ-loop superfamily (CDD accession No. cl21610) (Table 1), IbSWEET17 contained six transmembrane structures, the other eight IbSWEETs contained seven transmembrane domains (Figure 2), and the protein tertiary structure of IbSWEETs also clearly demonstrated the various transmembrane domains (Figure 3).

### 2.2. Protein Interaction Network of IbSWEETs

To explore the potential regulatory network of IbSWEETs, we constructed a SWEET interaction network based on *Arabidopsis* homologous proteins (Figure 4). Protein interaction prediction showed that seven SWEETs could interact with other proteins and SWEETs (SWEET1 and SWEET12; SWEET2 and SWEET17) could also interact with each other to form heterodimers. Except for SWEET7 and SWEET15, the other five SWEETs all interact with the sugar transporter protein (STP) or sucrose transporter (SUC). In addition, SWEET2 could interact with 26s proteasome subunit RPT1B; SWEET7 could interact with EPFL1, PILS (pins-likes), and transcription factors TCX7 and TCX8; and SWEET15 could interact with proteins related to plant senescence regulation SAG12/13, non-yellow coloring1 (NYC 1), transcription factors BHLH13/14, and NAC92. These results suggest that SWEETs are involved in the regulation of growth and development in sweet potato.

### 2.3. Sugar, Starch Level, and Weight of Sweet Potato Storage Roots

In order to understand the sugar, starch content, and individual weight changes of sweet potato storage roots, we determined the sugar and starch level and the individual weight of sweet potato storage roots at different periods (60, 90, and 120 days (d) after transplanting). As shown in Figure 5, the soluble sugar and starch content and the individual weight of single storage roots of sweet potato cultivars “xiguahong” (XGH) and “guijingshu 8” (GJ8) increased with the growth of storage roots. The soluble sugar content in XGH was higher than in GJ8 (Figure 5A) but the starch content was just the opposite (Figure 5B). The glucose and fructose content of XGH increased first and then decreased, while the glucose and fructose content of GJ8 did not change significantly and the content was lower than XGH in each period (Figure 5C). The sucrose content of both XGH and GJ8 continued to rise, and it was lower in GJ8 than XGH (Figure 5C). The individual weight of sweet potato storage roots was heavier in XGH than GJ8 at each period (Figure 5D).

### 2.4. Expression Patterns of IbSWEETs in Different Tissues and Developmental Periods

To understand the potential biological function of *IbSWEETs* in the growth and development of sweet potato, we analyzed the expression levels of nine *IbSWEETs* in the roots, stems, and leaves of two sweet potato cultivars (GJ8 and XGH) via real-time quantitative PCR (qRT-PCR) (Figure 6). Fibrous roots at 15 d after transplanting were used as a control to calculate the relative expression levels of roots, stems, and leaves at 30, 60, 90, and 120 d after transplanting. The results showed that the relative expression levels of *IbSWEET1a* and *IbSWEET17* were lower in roots, stems, and leaves (Figure 6) and the relative expression level of *IbSWEET2* was lower in roots and stems, while higher expression levels were detected in leaves, with about five times higher expression than in fibrous roots (Figure 6E,F); *IbSWEET1b* was mainly expressed in sweet potato roots and stably upregulated with the growth and development of the roots (Figure 6A,B); *IbSWEET15* was expressed at a low level in stems and was highly expressed by more than 15 times (Figure 6A,B) in the root expansion period (30~90 d after transplanting) and had the highest expression in leaves, up to 100 times (Figure 6E,F); *IbSWEET7*, *10a*, *10b*, and *12* were highly expressed in sweet potato leaves and more than 100 times that of roots and stems (Figure 6E,F). Nine *IbSWEET* genes shared the same expression trends in two sweet potato cultivars. These results suggest that *IbSWEETs* may play different roles in different developmental periods and various tissues of sweet potato.

To investigate the correlation between the sugar and starch content, individual root weight, and the expression of *IbSWEETs*, we analyzed the data using the bivariate Spearman of SPSS software (Version: 26). The results showed that the expression of *IbSWEET1b* was positively correlated with the soluble sugar, starch, sucrose content, and individual storage root weight; *IbSWEET15* expression was positively correlated with fructose content in XGH and GJ8. In addition, in XGH, the expression of *IbSWEET10b* was negatively correlated with the soluble sugar, starch, sucrose content, and individual storage root weight; the expression of *IbSWEET1a*, *7*, *10a*, and *12* was negatively correlated with glucose content, while *IbSWEET17* was opposite; the expression of *IbSWEET2* and *IbSWEET15* was positively correlated with fructose content (Table 2). In GJ8, the expression of *IbSWEET1b* and *IbSWEET7* was positively correlated with the soluble sugar, starch, sucrose content, and individual storage root weight and negatively correlated with fructose content, while the expression of *IbSWEET12* and *IbSWEET15* was negatively correlated with the above indexes but positively correlated with the fructose content. In addition, the expression of *IbSWEET1a* and *IbSWEET10a* was positively correlated with the glucose content (Table 3). These results indicate that the expression of *IbSWEETs* may be related to the sugar accumulation of sweet potato storage roots.

### 2.5. Expression Patterns of IbSWEETs in Response to Abiotic Stress

To further understand the physiological function of *IbSWEETs* under different abiotic stresses, we analyzed the expression patterns in leaves of sweet potato cultivar GJ8 after 4 °C, 200 mM NaCl, 20% PEG6000, 100 μM ABA treatment. The results showed that seven *IbSWEETs* showed different expression patterns (*IbSWEET1a* and *IbSWEET1b* were not expressed in leaves or the expression was extremely low) (Figure 7). After PEG treatment, the relative expression of *IbSWEET2*, *7*, and *15* decreased first and then increased compared to 0 h, *IbSWEET2* and *IbSWEET15* only increased near the expression level of treatment 0 h, while *IbSWEET7* increased to about seven times the expression level at 0 h. The relative expression levels of *IbSWEET10a*, *10b*, *12*, and *17* decreased after a slight increase, and they all fell below the 0 h expression level (Figure 7A).

After 4 °C treatment, the relative expression of *IbSWEET*2 did not change significantly with the extension of treatment time, that of *IbSWEET*7 was wavy, and the highest expression reached more than two times that of 0 h. The relative expression of *IbSWEET10a*, *12*, and *17* increased first and then decreased but was consistently higher than 0 h; that of *IbSWEET10b* increased steadily with the treatment time, while the relative expression of *IbSWEET15* decreased to around the expression level of 0 h (Figure 7B).

After NaCl treatment, *IbSWEET10a*, *10b*, and *12* had the highest expression at 1 h, then continuously decreased, and was less than 0 h. The relative expressions of *IbSWEET7* and *IbSWEET15* decreased first and then increased at 24 and 48 h. The relative expression of *IbSWEET17* was shown by a sustained rise and then a decrease below 0 h (Figure 7C).

After ABA treatment, the relative expression of *IbSWEET2* ascended after being slightly decreased and was maintained at about twofold expression. *IbSWEET10a*, *10b*, and *12* were all significantly induced at 1 h after treatment and then continued rising again after falling. The relative expression of *IbSWEET7* and *IbSWEET15* were lower than 0 h after ABA treatment, and that of *IbSWEET17* rose first and then decreased (Figure 7D).

### 2.6. Complementation of Yeast EBY.VW4000

SWEET has been demonstrated to mediate the transport of different sugars. To determine whether IbSWEETs transport sugars, IbSWEETs were heterologously expressed in the yeast mutant EBY.VW4000, with pDR196 as a negative control and AtSWEET1 as a positive control. The EBY.VW4000 transformed with empty vectors and constructs could be grown on synthetic deficient (SD) media containing 2% maltose, indicating the presence of the expression vector or target gene (Figure 8).

Expression of *IbSWEET7* effectively restored EBY.VW4000 growth on media supplemented with fructose, mannose, and glucose. Expression of *IbSWEET15* restored EBY.VW4000 growth on media containing glucose, and all yeast cells could grow slowly on galactose media.

2-deoxyglucose with poor metabolic capacity was used as a sensitive tool to detect the transport of sugar analogues [49]. The EBY.VW4000 transformed with the empty vectors were grown on medium supplemented with 2-deoxyglucose. All yeast cells, except those transformed with AtSWEET1, resumed growth on 1% maltose + 0.2% 2-deoxyglucose medium, indicating that these nine IbSWEETs can transport glucose analogues. In addition, IbSWEET10a, 10b, 12, and 15 could grow slowly in the medium supplemented with sucrose. The above results suggested that these nine IbSWEETs may play a role in hexose and sucrose transport (Figure 8).

## 3. Discussion

### 3.1. SWEET Mediates the Transport of Different Sugars

In 2010, Chen et al. [6] identified a new class of sugar-will-eventually-be-exported transporter (SWEET) proteins transporting glucose and other oligosaccharides from Arabidopsis by the fluorescence resonance energy transfer sensing technology (FRET sensor). It acts independently of H^+^-ATPase to transport sugar across the membrane [50,51,52]. Then, *SWEET* genes have been identified from many plants, such as rice, tomato, barley, alfalfa, garlic, etc. [52,53,54,55,56].

*AtSWEET1* has the ability to take up glucose in human embryonic kidney cells HEK293T and also has the same properties in yeast hexose absorption-deficient mutants (EBY.VW4000) and Xenopus oocytes [6,12]. The EBY.VW4000 mutant lacks the endogenous hexose transporters *HXT1*-*HXT17* and *GAL2* and grows normally on media containing maltose or slowly with galactose but not on media containing glucose, fructose, mannose, or sucrose [57]. AtSWEET8 is a glucose transport protein too and both AtSWEET8 and AtSWEET1 are abundantly expressed in Arabidopsis pollen tubes, suggesting that they may provide nutrients to the developing pollen tubes [6]. *AtSWEET11* and *AtSWEET12* mainly transport sucrose and, similarly, *OsSWEET11* and *OsSWEET14* are also transport carriers for low-affinity sucrose [12,13,58,59]. *AtSWEET17* express in parenchyma cells and vascular tissues; it is a vacuolar fructose transporter, mainly responsible for fructose transport in roots and leaves [60]; *AtSWEET16* also functions as a vacuolar sugar transporter responsible for glucose, fructose, and sucrose transport [61].

In this study, all nine IbSWEETs that heterologously expressed in yeast had glucose analogue transport activity but only two (IbSWEET7 and IbSWEET15) had glucose transport activity, while IbSWEET7 also had fructose and mannose transport activity. IbSWEET10a, 10b, 12, and 15 had weak sucrose transport activity and the other IbSWEETs did not detect sugar transport activity (Figure 8). In summary, different *SWEET* genes show differences in sugar transport activity.

### 3.2. SWEET Mediates Sugar Transport and Accumulation in Sweet Potato

The formation and development of sweet potato storage roots are crucial for root yield and quality. Storage root formation was thought to be a process of assimilate accumulation. Sucrose is the main product of photosynthesis, which is produced in mesophyll cells and then enters into the phloem cells for transport into various plant reservoir organs [62]. It has shown that *SWEETs* were involved in the transport and distribution of photosynthetic products in plants [12]. AtSWEET11 and AtSWEET12 localize on the plasma membrane of leaf phloem and are responsible for the transport of sucrose from parenchyma cells to the leaf exoplasm. The *atsweet11*, *12* double mutants exhibited slow-growing phenotype and reduced root sucrose content and increased leaf starch levels [12]. *ZmSWEET15a* had the highest expression in leaves and was highly correlated with leaf development and overexpression of *ZmSWEET15a* in maize reduced leaf sucrose content and increased grain sucrose content [63]. *IbSWEET10a*, *10b*, *12*, and *15* were highly expressed in leaves (Figure 6E,F) and had weak sucrose transport activity (Figure 8), so they may be involved in the export of sugar from sweet potato leaves.

In barley, reducing *HvSWEET11b* expression in developing grains reduced grain size, sink strength, endopolyploid endosperm cell number, and starch and protein content [64]. The highest expression of *SlSWEET12c* was observed in the red ripening stage of tomato fruit. Silencing of *SlSWEET12c* increased sucrose accumulation and reduced hexose quantity in tomato fruit; the opposite effect was observed with overexpression of *SlSWEET12c* [18]. Similarly, silencing of *SlSWEET7a* or *SlSWEET14* increased the sugar content of the mature fruit, resulting in higher plants and larger fruit [31]. In watermelon, *ClSWEET3* was involved in the storage of plasma membrane sugar transport in the vacuole of fruit cells. Knockdown of *ClSWEET3* affected the sugar accumulation in the fruits [65]. In citrus, *CitSWEET11d* transcript was significantly correlated with sucrose accumulation, and sucrose levels of overexpressing *CitSWEET11d* in citrus calli and tomato fruits were higher than in wild type, indicating that *CitSWEET11d* promoted sucrose accumulation [66].

During the expansion of sweet potato storage roots, the content of sucrose and soluble sugar continued to increase (Figure 5) and the transcript of *IbSWEET1b* showed a significant positive correlation with the soluble sugar and sucrose of the storage roots (Table 2 and Table 3). Further comparison of *IbSWEET1b* expression in different tissues and at different developmental stages also suggested that *IbSWEET1b* was highly expressed in storage roots, indicating that it may play a role in sweet potato storage roots. *IbSWEET15* also had high expression in the stage of storage root expansion and had glucose and sucrose transport activity, indicating that this gene plays a certain role in the swelling stage of sweet potato. In general, *IbSWEET1b* and *IbSWEET15* may mediate the sugar transport and accumulation of the storage root of sweet potato.

### 3.3. SWEETs Participate in Regulating Abiotic Stress in Sweet Potato

Sucrose is the main product of photosynthesis and it is also penetrant, which prevents tissue damage under limited water availability. Thus, sucrose transport and distribution are key processes in plants in response to abiotic stresses [35]. Sugar concentrations increased significantly under biotic and abiotic stresses, such as cold, drought, and pathogen challenge [67,68]. The *SWEET* genes were reported to respond to various abiotic stresses. Under drought stress, *AtSWEET4*, *11*, *12*, *13*, *14*, and *15*, *OsSWEET12*, *15*, and *16* were induced [33,69]; after treatment with polyethylene glycol (PEG), *AtSWEET2*, *11*, *13*, and *15* were downregulated and sucrose transport between leaves and roots were decreased [7]. *JcSWEET16* responded to drought and salt stress in leaves and the overexpression of *JcSWEET16* in Arabidopsis altered the flowering time and salt tolerance levels without affecting the drought tolerance [70]. *AtSWEET17* was a fructose-specific sugar transporter localized in the vacuolar membrane and the knockout mutant lines of *AtSWEET17* exhibited reductive LR growth and reduced expression of LR development-associated transcription factors under drought stress, resulting in impaired drought tolerance in *atsweet17* mutant lines [37].

In tea plants, *CsSWEET* genes played an important role in response to abiotic stress. *CsSWEET16* contributed to sugar compartmentalization in the vacuole and to modifying cold tolerance in Arabidopsis [71]. Overexpression of *MdSWEET16* in apple calli reduced its sucrose content but increased its cold tolerance [72] and transgenic plants overexpressing *AtSWEET4* showed high freezing tolerance too [73]. Similarly, *HfSWEET17* was overexpressed in tobacco and transgenic plants showed significantly more cold resistance than wild-type tobacco [34].

*ZmSWEET1b* was a typical sugar transporter, and the *ZmSWEET1b* knockout lines had significantly reduced sucrose and fructose content. After salt treatment, the *ZmSWEET1b* edited lines became more withered [74], while overexpression of *DsSWEET17* improved salt tolerance in Arabidopsis [75]. The ABA-responsive transcription factor *OsbZIP72* binded to the promoter, inducing high expression of *OsSWEET13* and *OsSWEET15* and regulating the transport and distribution of sucrose under abiotic stress, and this mechanism may be targeted for maintaining rice sugar homeostasis under drought and salt stress [35].

Moreover, some genes confer multiple stress tolerance to plants; for example, *DsSWEET17* transgenic Arabidopsis had high tolerance to salt, osmotic, and oxidative stresses [76]. In our study, the *SWEET* genes had differential responses under various abiotic stresses. Under PEG treatment, *IbSWEET2* and *IbSWEET7* were upregulated and *IbSWEET10a*, *10b*, *12*, and *15* were downregulated. Under 4 °C treatment, *IbSWEET10a*, *10b*, *12*, and *17* were induced and the expression was upregulated. After NaCl treatment, *IbSWEET10a*, *10b*, and *12* were strongly induced at 1 h after treatment, and *IbSWEET2*,*10a*, *10b*, *12*, and *17* were also induced after ABA treatment (Figure 8), suggesting that *IbSWEETs* may regulate the response of sweet potato to abiotic stress by regulating sugar transport and distribution.

## 4. Materials and Methods

### 4.1. Plant Materials and Growth Conditions

The sweet potato (*Ipomoea batatas* (L.) Lam) cultivars “xiguahong” (XGH) and “guijingshu 8” (GJ8) were used in this experiment. In early September, 30cm long stem tips were planted in the farm of Guangxi University (108°22′ E, 22°48′ N). Fibrous roots were collected at 15 days after transplanting; mature leaves (+2–+3 leaves) and corresponding stems and roots were collected at 30, 60, 90, and 120 days, respectively. The GJ8 25cm long stem tips growing consistently at 60 days (vigorous growth period) after cutting were taken and cultured in Hoagland nutrient solution until the roots grew to about 10 cm, processing them with 200 mM NaCl, 20% PEG6000, 100 μM ABA, and low temperature (4 °C). The third expanded leaves were collected at 0, 1, 6, 12, 24, and 48 h after NaCl, PEG, and ABA treatments and at 0, 1, 3, 6, and 12 h after 4 °C treatment, with three biological replicates. Samples were frozen immediately in liquid nitrogen and kept at −80 °C until analysis.

A part of storage roots harvested at 60, 90, and 120 days after transplanting were sliced then put into the oven at 105 °C and, after 30 min, the temperature was set to 70 °C and held until the samples were dried. Samples were crushed and screened for determining the soluble sugar content and starch content.

### 4.2. Gene Sequence and Phylogenetic Analysis

The molecular weight (MW), protein isoelectric point (pI), instability index, aliphatic index and hydrophilicity (GRAVY) of the IbSWEETs were analyzed by ExPASy (https://web.expasy.org/protparam/ (accessed on 13 November 2023)) [77]. Subcellular localization and protein tertiary structure prediction were analyzed via WoLF PSORT (https://www.genscript.com/wolf-psort.html (accessed on 13 November 2023)) and SWISS-MODEL (https://swissmodel.expasy.org/ (accessed on 13 November 2023)) [78].

The SWEET proteins from sweet potato and *Arabidopsis thaliana*, rice, and grape were used to establish the phylogenetic tree by MEGA X [79] with the neighbor-joining (NJ) method using the default settings, and the results were displayed with iTOL (http://itol.embl.de/ (accessed on 13 November 2023)).

### 4.3. SWEET Protein Interaction Network Prediction

Based on *Arabidopsis* homologue proteins, the SWEET protein interaction network was predicted using STRING (Version: 12.0, https://cn.string-db.org/ (accessed on 13 November 2023)) [80] and the protein interaction network was drawn using Cytoscape software (Version: 3.9.1) [81].

### 4.4. RNA Extraction and qRT-PCR

Total RNA was extracted using a plant RNA fast pickup kit (single column type) (Magen, code R4014-02, Guangzhou City, China) and the first strand cDNA synthesis was performed using TaKaRa PrimeScript RT reagent Kit with gDNA Eraser (Perfect Real Time) (TaKaRa, code RR047, Kyoto, Japan). The experimental operation was performed according to the kit instructions. The RNA concentration was obtained by measuring the absorption value of OD_260_ by ultramicrospectrophotometer (Themo Scientific NANO DROP ONE, Waltham, MA, USA). In a 20 μL reaction system, the first strand of cDNA was synthesized using 1000 ng RNA as a template. All qRT-PCR was performed in BIO-RAD CFX96 real-time system. The expression level was calculated using the 2^−ΔΔCT^ method with *IbACTIN* gene as an internal reference.

### 4.5. Determination of Sugar and Starch

Soluble sugar and starch content were determined with the method described by Gao et al. [82]. Glucose, fructose, and sucrose were determined by ion chromatography. Chromatographic conditions: Dionex CarboPac PA1 Anion-exchange column, including analytical column (2 × 250 mm) and protective column (2 × 250 mm); column temperature was 35 °C. Mobile phase A was ultrapure water, B was NaOH aqueous solution (200 mmol), A: B was 91%: 9%, flow rate was 1 mL/min, and sample intake was 25 μL.

### 4.6. Functional Characterization of IbSWEETs by Heterologous Expression in Yeast

To test the biochemical properties of IbSWEETs, we constructed plasmids with an *IbSWEETs*-coding sequence domain in the yeast expression vector pDR196. The recombinant plasmids were obtained by inserting the coding sequences of *IbSWEETs* and *AtSWEET1* (positive control) between pDR196 *sal*I and the *xho*I restriction site (Appendix A).

The recombinant plasmid and pDR196 transformed hexose uptake-deficient yeast EBY.VW4000; the transformants were grown in synthetic deficient (SD) medium containing 2% (*w*/*v*) maltose as the sole carbon source and shaken to OD_600_ 0.6 at 30 °C, 180 rpm. The bacterial solution was diluted to 10 times, 100 times, and 1000 times, and 10 μL diluted droplets were placed in SD/_-uracil_ solid medium with 2% (*w*/*v*) maltose, glucose, sucrose, mannose, galactose, fructose, and 1% maltose + 0.2% 2-deoxyglucose as the sole carbon source, respectively. Yeast cells were grown for 3 to 5 days at 30 °C. Cell growth on different media were observed to study in which sugars that IbSWEETs proteins were involved.

## 5. Conclusions

In this study, nine *IbSWEET* genes were obtained, all with 2 MtN3 domains or PQ-loop superfamily and six to seven transmembrane domains and belonging to four different clades. Protein interaction prediction showed that SWEETs interact with monosaccharide transporters, sucrose transporters, and some transcription factors. qRT-PCR analysis demonstrated obvious tissue specificity of *IbSWEETs* and large differences in response to abiotic stresses. The expression level of some *IbSWEETs* showed a significant positive/negative relationship with the sugar and starch content and the weight of sweet potato storage roots. Yeast heterologous expression suggested that IbSWEETs transported different classes of sugars. The nine SWEET genes have different lengths, physicochemical properties, and large differences in protein tertiary structure. These differences may lead to their different sugar transport activities, so their response to abiotic stress was different and the correlation of sugar and starch accumulation between sweet potatoes was also different. The results provide a basis for further clarifying the function of SWEET genes in the growth and development of sweet potato.

## Figures and Tables

**Figure 1 ijms-24-16615-f001:**
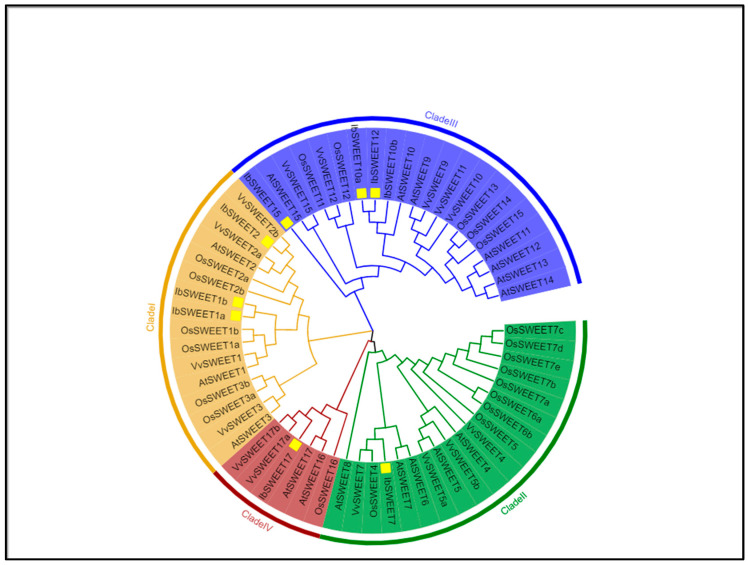
Phylogenetic relationships of 9 IbSWEETs and 17 AtSWEETs, 21 OsSWEETs, and 15 VvSWEETs. The neighbor-joining tree was generated using MEGA X with 1000 bootstrap replicates. *SWEET* gene family are distinguished by different colors. The yellow square marks are sweet potato SWEETs.

**Figure 2 ijms-24-16615-f002:**
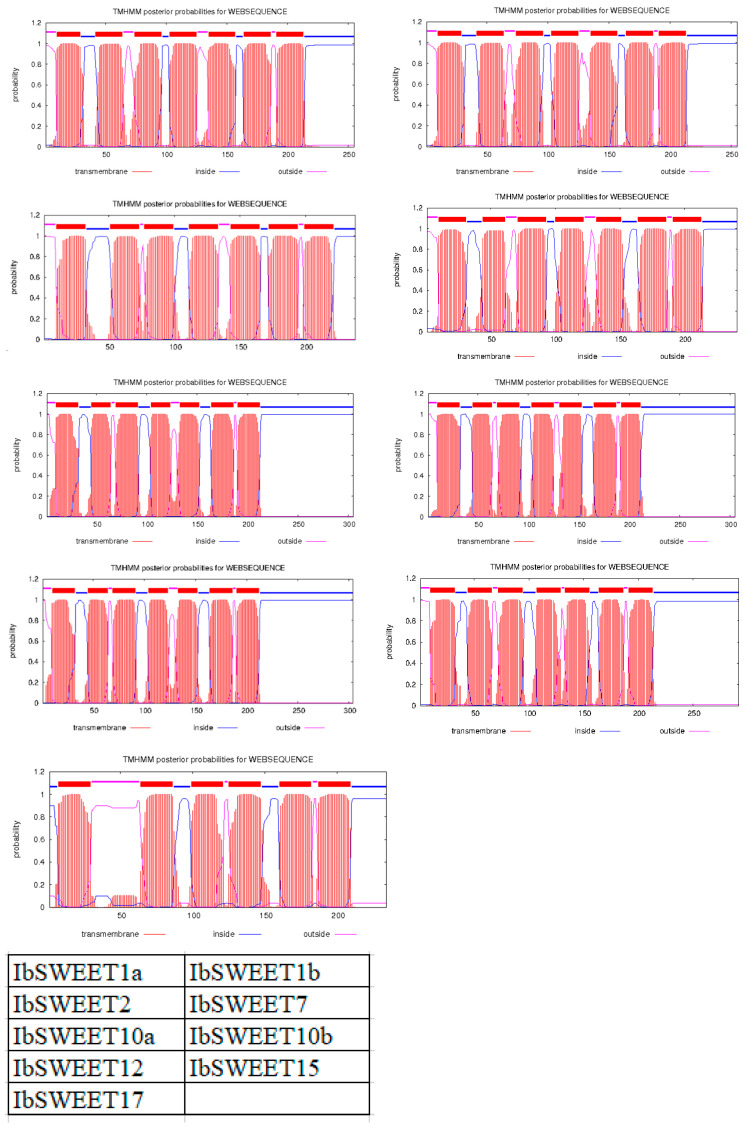
The transmembrane domains of IbSWEET proteins. The position of N- and C-terminal domains of the protein are indicated by pink or blue lines. Table on the right bottom shows the location of each protein in this figure.

**Figure 3 ijms-24-16615-f003:**
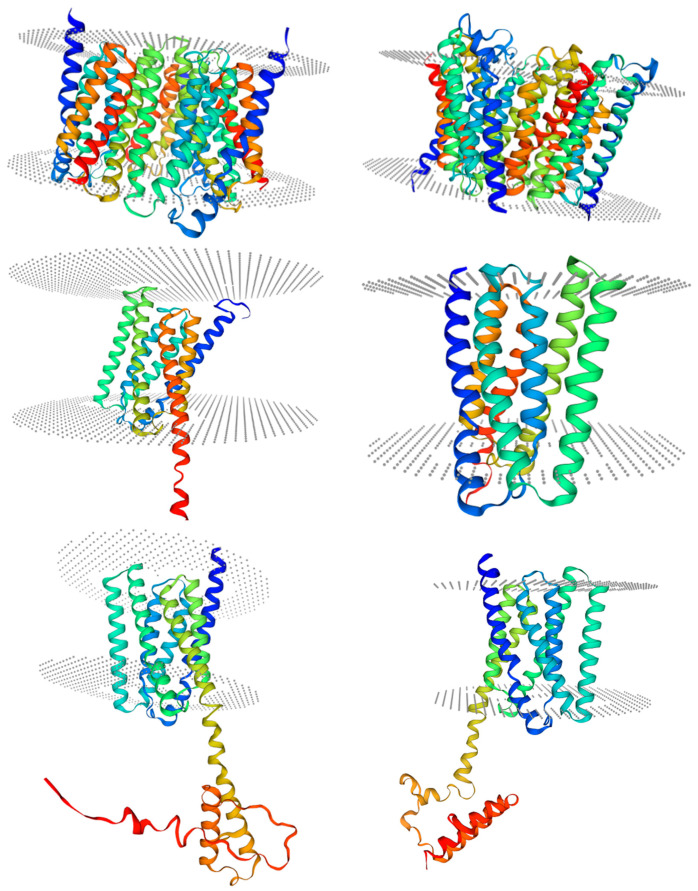
Structural modeling of IbSWEETs protein. Gray dots indicate the phospholipid bilayer. Table on the right bottom shows the location of each protein in this figure.

**Figure 4 ijms-24-16615-f004:**
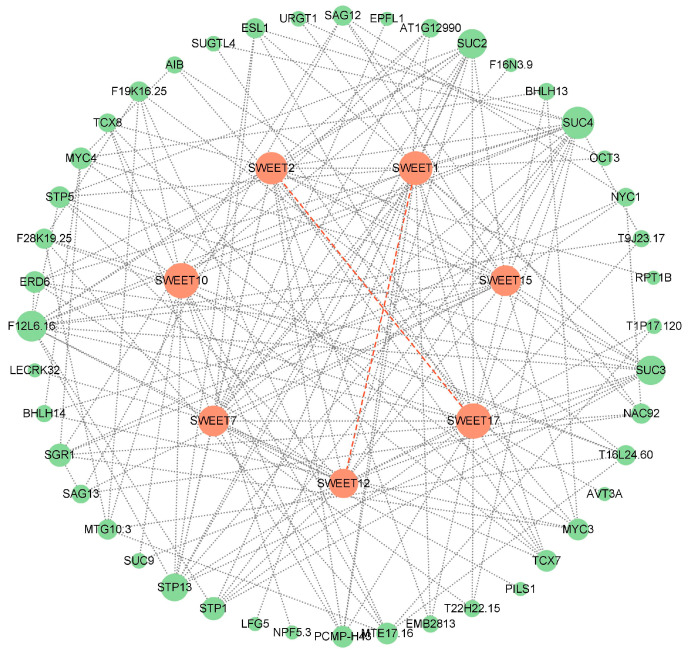
Functional interaction networks of IbSWEETs in sweet potato according to orthologues in Arabidopsis. Network nodes represent proteins and lines indicate protein–protein associations. The size of the nodes indicate the number of interacting proteins. Orange circles represent SWEET proteins and green circles represent other proteins; the dashed orange line represents the formation of heterodimers between two SWEETs.

**Figure 5 ijms-24-16615-f005:**
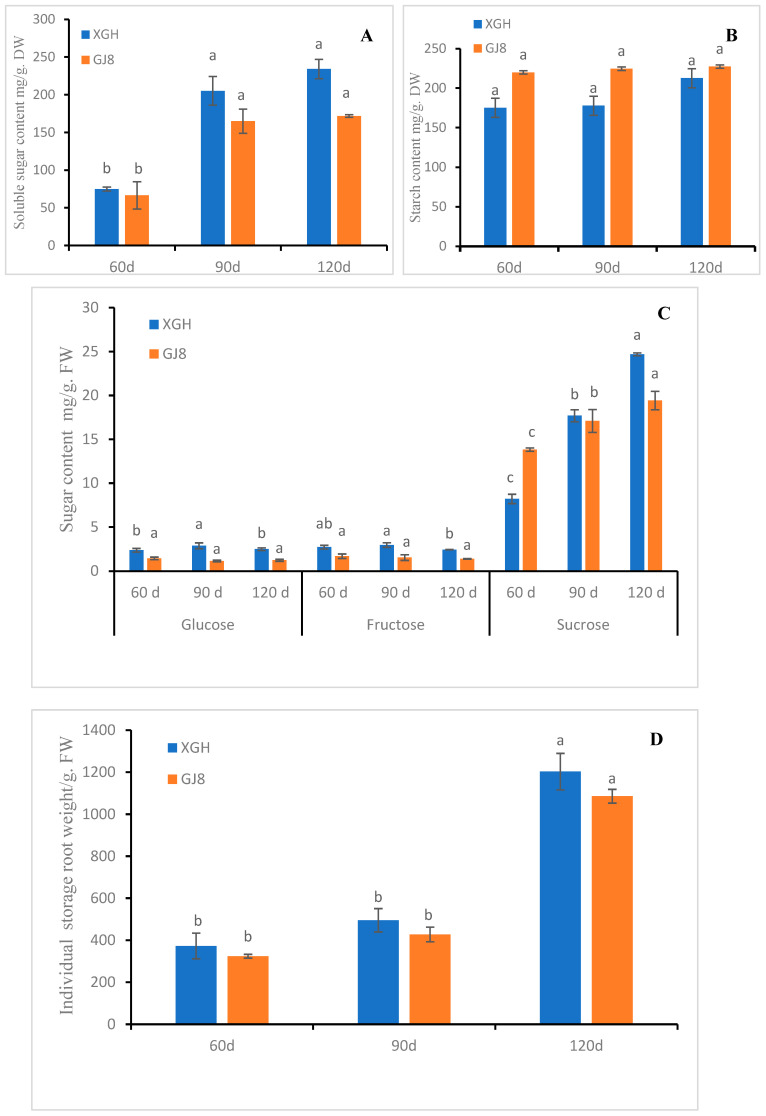
Sugar, starch content, and weight of storage root changes in two sweet potato cultivars in different periods. (**A**): soluble sugar content. (**B**): starch content. (**C**): glucose, fructose, and sucrose content. (**D**): individual storage root weight. Letters ‘a–c’ indicate different changes in different development periods of the same cultivar (*p* < 0.05), according to Duncan’s multiple range test using SPSS software (Version: 26).

**Figure 6 ijms-24-16615-f006:**
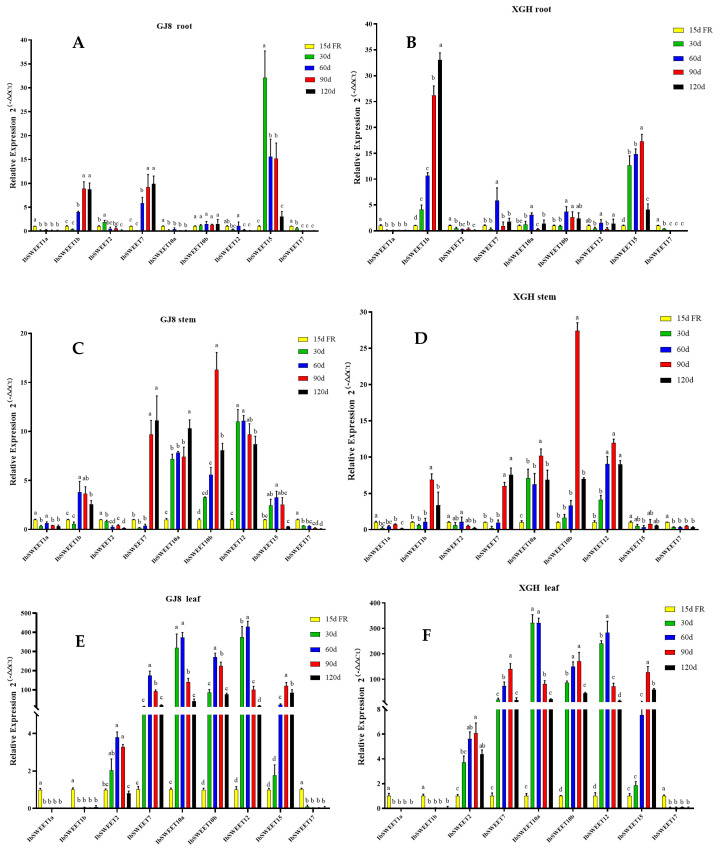
*IbSWEETs* expression analysis of different sites at two sweet potato cultivars in different periods. Note: the left side is the expression analysis in the root (**A**), stem (**C**), and leaf (**E**) of GJ8, respectively. The right side is the expression analysis in the root (**B**), stem (**D**), and leaf (**F**) of XGH, respectively. FR: fibrous root. Letters ‘a–d’ indicate different changes in different development periods of the same *SWEET* gene (*p* < 0.05), according to Duncan’s multiple range test using SPSS software (Version: 26).

**Figure 7 ijms-24-16615-f007:**
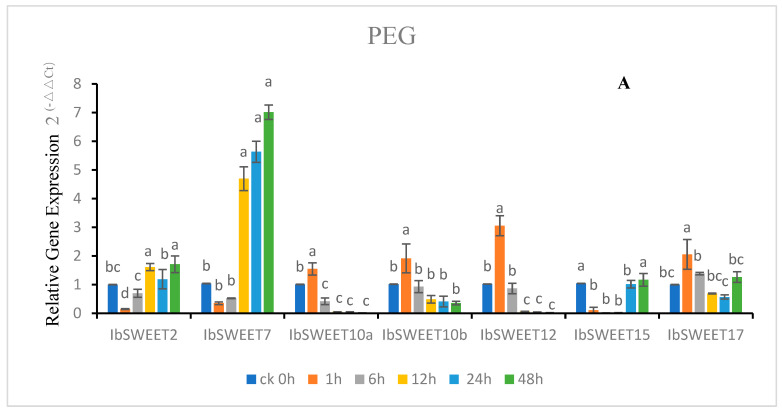
Gene expression patterns of *IbSWEETs* in response to different abiotic stresses in GJ8 as determined by qRT-PCR. Note: (**A**) 20% PEG6000, (**B**) 4 °C, (**C**) 200 mM NaCl, (**D**) 100 μM ABA. Letters ‘a–d’ indicate different changes in different treatment periods of the same *SWEET* gene (*p* < 0.05), according to Duncan’s multiple range test using SPSS software (Version: 26).

**Figure 8 ijms-24-16615-f008:**
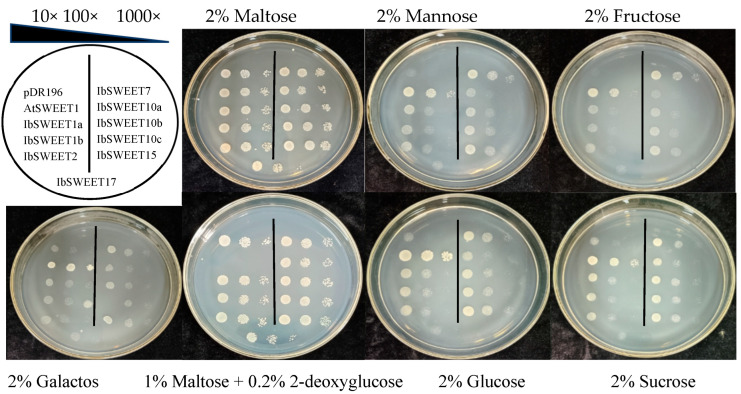
Complementation growth assay in the yeast EBY.VW4000 mutant.

**Table 1 ijms-24-16615-t001:** The protein characterization of 9 IbSWEETs in sweet potato.

Gene Name	Number of Amino Acids	Molecular Weight (kD)	Theoretical pI	MtN3/Saliva (PQ-Loop Repeat)	Instability Index (Ⅱ)	Aliphatic Index	GRAVY	Subcellular Localization
*IbSWEET1a*	255	28.41	9.44	2	50.47	114.63	0.775	plas
*IbSWEET1b*	255	28.58	9.51	2	47.72	115.41	0.779	plas
*IbSWEET2*	238	26.38	9.11	2	51.82	122.82	0.947	chlo
*IbSWEET7*	241	26.72	9	2	24.29	126.97	0.921	chlo
*IbSWEET10a*	305	34.36	9.27	2	36.61	117.48	0.614	chlo
*IbSWEET10b*	305	34.07	9.34	2	41.88	114.72	0.549	chlo
*IbSWEET12*	304	34.28	9.19	2	39.42	116.28	0.58	chlo
*IbSWEET15*	292	32.8	7.01	2	42.48	120.17	0.684	chlo
*IbSWEET17*	234	25.61	5.46	2	37.97	120.81	0.859	vacu

Note: chlo: chloroplast; plas: plasma membrane; vacu: vacuole.

**Table 2 ijms-24-16615-t002:** Correlation between expression of *IbSWEETs* and sugar, starch content, and storage root weight of XGH.

Index	IbSWEET1a	IbSWEET1b	IbSWEET2	IbSWEET7	IbSWEET10a	IbSWEET10b	IbSWEET12	IbSWEET15	IbSWEET17
Soluble sugar		r = 1, *p* < 0.01				r = −1, *p* < 0.01			
Starch		r = 1, *p* < 0.01				r = −1, *p* < 0.01			
Glucose	r = −1, *p* < 0.01			r = −1, *p* < 0.01	r = −1, *p* < 0.01		r = −1, *p* < 0.01		r = 1, *p* < 0.01
Fructose			r = 1, *p* < 0.01					r = 1, *p* < 0.01	
Sucrose		r = 1, *p* < 0.01				r = −1, *p* < 0.01			
Storage weight		r = 1, *p* < 0.01				r = −1, *p* < 0.01			

**Table 3 ijms-24-16615-t003:** Correlation between expression of *IbSWEETs* and sugar, starch content, and storage root weight of GJ8.

Index	IbSWEET1a	IbSWEET1b	IbSWEET2	IbSWEET7	IbSWEET10a	IbSWEET10b	IbSWEET12	IbSWEET15	IbSWEET17
Soluble sugar		r = 1, *p* < 0.01		r = 1, *p* < 0.01			r = −1, *p* < 0.01	r = −1, *p* < 0.01	
Starch		r = 1, *p* < 0.01		r = 1, *p* < 0.01			r = −1, *p* < 0.01	r = −1, *p* < 0.01	
Glucose	r = 1, *p* < 0.01				r = 1, *p* < 0.01				
Fructose		r = −1, *p* < 0.01		r = −1, *p* < 0.01			r = 1, *p* < 0.01	r = 1, *p* < 0.01	
Sucrose		r = 1, *p* < 0.01		r = 1, *p* < 0.01			r = −1, *p* < 0.01	r = −1, *p* < 0.01	
Storage weight		r = 1, *p* < 0.01		r = 1, *p* < 0.01			r = −1, *p* < 0.01	r = −1, *p* < 0.01	

## Data Availability

The data generated and analyzed during the present study are available on request from the corresponding author.

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
