# Peer review of "Molecular Cloning, Expression Analysis, and Functional Analysis of Nine IbSWEETs in Ipomoea batatas (L.) Lam"

_ijms, 2023, doi:10.3390/ijms242316615_

Round 1

Reviewer 1 Report

Comments and Suggestions for Authors

This paper deals with the investigation of sweet potato genes encoding sugar transporters. The authors showed the function of those proteins and the results suggested the importance in the sink-source dynamics. I have a few minor comments.

1) In abstract, I am glad if the authors describe the general function of SWEET protein. The readers will have interests in the protein.

2) In introduction, the authors should explain the situation of genetic resources in sweet potato and the reason why the authors performed the cDNA screening. I understand the difficult situation in sweet potato as reviewed by https://doi.org/10.1016/j.xplc.2022.100332.

3) Other sections are fine.

Comments on the Quality of English Language

Line 75, the sentence seems to be grammatically incorrect. "Yeast heterologous expression analysis the species of sugars 75 transported by IbSWEETs."

Author Response

Dear reviewer:

Thank you very much for taking the time to review this manuscript entitled “Molecular cloning, expression analysis and functional analysis of 9 IbSWEETs in Ipomoea batatas (L.) Lam” (ID: ijms-2674942). Those comments are all valuable and very helpful for revising and improving our paper, as well as the important guiding significance to our researches. We have studied comments carefully and have made correction which we hope meet with approval. Revised portion are marked in red in the paper.

3. Point-by-point response to Comments and Suggestions for Authors

Comments 1: In abstract, I am glad if the authors describe the general function of SWEET protein. The readers will have interests in the protein.

Response 1: Thanks so much for your comments. The functions of stress response and sugar metabolism of SWEET protein had been added.

Comments 2: In introduction, the authors should explain the situation of genetic resources in sweet potato and the reason why the authors performed the cDNA screening. I understand the difficult situation in sweet potato as reviewed by https://doi.org/10.1016/j.xplc.2022.100332.

Response 2: Thank you for pointing this out. We agree with this comment. Therefore, we have added this content as flowing: (1) Sweet potato is a highly heterozygous hexaploid plant, so the genetic study of sweet potato is very complicated. The development of genetics and genomics has enabled sweet potato and crop improvement to facilitate and accelerate genetic improvement of this important rhizome crop through breeding combining state-of-the-art multi-genomics such as genomic selection and gene editing. Agrobacterium tumefaciens- mediated calli suspension culture transformation of sweet potato provides the possibility of genome - modified breeding (including transgenic and genome editing breeding). (2) Given the important role of SWEETs in plant development, and to gain a more comprehensive and deeper understanding of the role of SWEET genes in the growth, development and sugar accumulation of sweet potato, we did further research. In the previous study, nine SWEETs were screened for up- / downregulated by RNA-seq datasets (submitted to NCBI with the accession number PRJNA678375) and obtained the SWEET gene sequences from the sweet potato database (http://public-genomesngs.molgen.mpg.de/SweetPotato/). This change can be found in lines 74-80 and lines 86-91 in the revised manuscript.

Once again, thank you very much for your comments and suggestions.

Reviewer 2 Report

Comments and Suggestions for Authors

This study identifies nine IbSWEET genes in sweet potato, which belong to four clades and interact with other proteins to form a heterodimer. These genes are tissue-specific, responsive to abiotic stresses, and can transport different sugar species, providing insights for further understanding SWEET gene functions and molecular breeding. The reviewer appreciates the effort of the authors to prove their hypothesis using series experiments. However, the reviewer has a few major comments regarding this study. Thus, the authors need to consider the following comments to improve the quality of this manuscript.

In section 2.1. There is no information about molecular cloning. Check it.

The materials and methods section looks shallow and seems it was hastily written. Especially subsections 4.3 and 4.4.

References are not properly cited in the appropriate places in the materials and methods section.

In section 4.3: Authors are advised to provide the Cytoscape software version and cite the appropriate references for both STRING and Cytoscape. Written information is not enough for the MM section.

Please describe more about in the RNA extraction and qRT-PCR section in the materials and method section. What is RNA concentration? How much concentration was used for cDNA conversion. Cite the appropriate reference.

Authors should provide the vector construction and molecular cloning image in the supplementary section for ease of understanding of the readers.

Authors are advised to provide gel image for confirmation of cloning with restriction digestion in supplementary section or peer review purposes.

Gene names should be in italics. Check and revise the same throughout the manuscript. Eg. lines 18, 263, etc.

Line 19: responsive/ responsible to abiotic stresses.

Authors are advised to fix grammar, space and punctuation errors throughout the manuscript. Because it seems the manuscript was hastily written.

Line 367: Hoogland or Hoagland? Correct it.

Line 263: AtSWEET1, italics?

In figure 7, write either “relative fold change in gene expression or Relative gene expression”.

Line 371: space error.

Lines 372-374: Syntax error. Reframe it.

In section 4.2: seems no starting point.

Line 381: Scientific name (Arabidopsis thaliana) should be italics.

Line 396: anthracene-sulfuric acid method (reference)?

Appropriate reference should be cited for MEGA X tool.

Line 406: sal and xho should be italics.

If possible, please perform gene ontology enrichment analysis of 9 IbSWEETs genes.

In conclusion section, write a few lines about future perspectives or hypotheses about the study. It will be useful to the readers for ease of understanding to design their study related to this studied issue.

Comments on the Quality of English Language

Authors are advised to fix grammar, space and punctuation errors throughout the manuscript

Author Response

Dear Reviewer:

Thank you very much for taking the time to review this manuscript entitled “Molecular cloning, expression analysis and functional analysis of 9 IbSWEETs in Ipomoea batatas (L.) Lam” (ID: ijms-2674942). Those comments are all valuable and very helpful for revising and improving our paper, as well as the important guiding significance to our researches. We have studied comments carefully and have made correction which we hope meet with approval. Revised portion are marked in red in the paper.

2. Questions for General Evaluation

Reviewer’s Evaluation

Response and Revisions

Does the introduction provide sufficient background and include all relevant references?

Yes

Are all the cited references relevant to the research?

Must be improved

We have checked and modified it.

Is the research design appropriate?

Can be improved

Are the methods adequately described?

Must be improved

We have already modified it.

Are the results clearly presented?

Can be improved

Are the conclusions supported by the results?

Can be improved

3. Point-by-point response to Comments and Suggestions for Authors

Comments 1: In section 2.1. There is no information about molecular cloning. Check it.

Response 1: Thank you. We have added gel image of the SWEET genes cloning in Supplementary data. This change can be found in line 102 in the revised manuscript.

Comments 2: The materials and methods section looks shallow and seems it was hastily written. Especially subsections 4.3 and 4.4.

Response 2: Thank you. We have refined the subsections 4.3 and 4.4 to make it more detailed, and cited relevant references. This change can be found in lines 414-423 in the revised manuscript.

Comments 3: References are not properly cited in the appropriate places in the materials and methods section.

Response 3: Thank you. We have checked all references in the materials and methods section to ensure that the references were relevant to the content. This change can be found in lines 409, 414, 427 in the revised manuscript.

Comments 4: In section 4.3: Authors are advised to provide the Cytoscape software version and cite the appropriate references for both STRING and Cytoscape. Written information is not enough for the MM section.

Response 4: Thank you . We have added the Cytoscape and STRING software version, and cited the appropriate references for both STRING and Cytoscape. This change can be found in lines 414-415 in the revised manuscript.

Comments 5: Please describe more about in the RNA extraction and qRT-PCR section in the materials and method section. What is RNA concentration? How much concentration was used for cDNA conversion. Cite the appropriate reference.

Response 5: Thank you. The experimental operation was performed according to the kit instructions. The RNA concentration was obtained by measuring the absorption value of OD260 by ultramicrospectrophotometer (themo scientific NANO DROP ONE, USA). 1000 ng RNA was used with a final volume of 20 μL for the first strand cDNA synthesis. This change can be found in lines 420-423 in the revised manuscript.

Comments 6: Authors should provide the vector construction and molecular cloning image in the supplementary section for ease of understanding of the readers.

Response 6: Thank you. We have added the vector construction and molecular cloning image in the supplementary section. This change can be found in supplementary data and in lines 437-438 in the revised manuscript.

Comments 7: Authors are advised to provide gel image for confirmation of cloning with restriction digestion in supplementary section or peer review purposes.

Response 7: Thank you. We have added the molecular cloning gel image and the vector construction image in the supplementary section. This change can be found in supplementary data.

Comments 8: Gene names should be in italics. Check and revise the same throughout the manuscript. Eg. lines 18, 263, etc.

Response 8: Thank you. We have examined the entire manuscript and modified the gene name as italic. This revise can be found in lines 286 in the revised manuscript.

Comments 9: Line 19: responsive/ responsible to abiotic stresses.

Response 9: Thank you. We have re-written this part according to the Reviewer’s suggestion. This change can be found in lines 17-29 in the revised manuscript.

Comments 10: Authors are advised to fix grammar, space and punctuation errors throughout the manuscript. Because it seems the manuscript was hastily written.

Response 10: Thank you. We have carefully examined the manuscript and revised the fix grammar, space and punctuation errors. This revise can be found in lines 58,59,63,84,133,200,248,254,392,398-401,403,408,428,437 in the revised manuscript.

Comments 11: Line 367: Hoogland or Hoagland? Correct it.

Response 11: Thank you. It should be “Hoagland”, and has been corrected. This revise can be found in line 392 in the revised manuscript.

Comments 12: Line 263: AtSWEET1, italics?

Response 12: Yes, we have corrected it. This revise can be found in line 286 in the revised manuscript.

Comments 13: In figure 7, write either “relative fold change in gene expression or Relative gene expression”.

Response 13: Thank you. We have modified it as “Relative gene expression patterns of IbSWEETs in response to different abiotic stress in GJ8 as determined by qRT - PCR”. This revise can be found in line 254 in the revised manuscript.

Comments 14: Line 371: space error.

Response 14: Thank you. We have re-written this section. This revise can be found in lines 396-397 in the revised manuscript.

Comments 15: Lines 372-374: Syntax error. Reframe it.

Response 15: Thank you. We have re-written this section. This revise can be found in lines 398-401 in the revised manuscript.

Comments 16: In section 4.2: seems no starting point.

Response 16: Thank you. We have modified it as “The molecular weight (MW), protein isoelectric point (pI), instability index, aliphatic index and hydrophilicity (GRAVY) of the IbSWEETs were analyzed by ExPASy.” This revise can be found in lines 403-404 in the revised manuscript.

Comments 17: Line 381: Scientific name (Arabidopsis thaliana) should be italics.

Response 17: Thank you. It has been corrected. This revise can be found in line 408 in the revised manuscript.

Comments 18: Line 396: anthracene-sulfuric acid method (reference)?

Response 18: Thank you. We referred to the method described by Gao et al. (2022) and cited the relevant references. This changed can be found in line 427 in the revised manuscript.

Comments 19: Appropriate reference should be cited for MEGA X tool.

Response 19: Thank you . We have cited the relevant references. This changed can be found in line 409 in the revised manuscript.

Comments 20: Line 406: sal and xho should be italics.

Response 20: Thank you. It has been corrected. This revise can be found in line 437 in the revised manuscript.

Comments 21: If possible, please perform gene ontology enrichment analysis of 9 IbSWEETs genes.

Response 21: Thank you. The advice is very good, but we’re going to put this content in a follow-up paper.

Comments 22: In conclusion section, write a few lines about future perspectives or hypotheses about the study. It will be useful to the readers for ease of understanding to design their study related to this studied issue.

Response 22: Thank you. We have added the content: The nine SWEET genes have different lengths, physicochemical properties and large differences in protein tertiary structure. Perhaps these differences lead to their different sugar transport activities, so their response to abiotic stress was different, and the correlation of sugar and starch accumulation in sweet potato was also different. This changed can be found in lines 457-460 in the revised manuscript.

Once again, thank you very much for your comments and suggestions.

Reviewer 3 Report

Comments and Suggestions for Authors

Comments for the manuscript entitled: "Molecular cloning, expression analysis and functional analysis of 9 IbSWEETs in Ipomoea batatas" submitted by Jingli Huang et al. 

This study is interesting, rich in data, about SWEET genes and SWEET proteins from Ipomoea batatas. 9 SWEET genes are highlighted in the sweet potato (IbSWEET1a, 1b, 2, 7, 10a, 10b, 12 and 17) obtained by PCR amplification from the cDNA of the sweet potato.

The study focuses on the characterization and especially the role of IbSWEET proteins, taking into account that there are few reports about them.

The expression levels of IbSWEET genes were monitored in two cultivars of sweet potato, in three organs (roots, stems, leaves), in different periods of development. In this sense, it has been shown that the accumulation of sugar in the roots of sweet potato is related to the expression IbSWEETs.

It has been demonstrated that IbSWEETs expression was different in response to abiotic stresses caused by PEG, 4 oC, NaCl, ABA treatments.

Also, the 9 IbSWEETs have a role in the transport of hexoze and sucrose, a fact proven by the heterologous expression of IbSWEETs in the yeast mutant EBY.VW4000.

The obtained and argued results help to understand (largely) the involvement of SWEET genes in the processes of sweet potato growth and development, as well as promoting molecular reproduction in sweet potato.

My comments are below:

1. In the title of the manuscript, the scientific name should be complete, ie: Ipomoea batatas (L.) Lam

2. Given the multitude of data, the Abstract should be more consistent.

3. The objectives and the aim of the research are not clearly outlined.

4. In line 27, you probably wanted the phrase to start with "Sucrose, as the main ..". Otherwise, the phrase would be too long!

5. In line 50, Mainly should be mainly. The next sentence to begin with: Loss of AtSWEET13 and ...

6. In lines 54-55, you omited the verb: "In addition, SISWEET5b [25],....and so on are also associated with flower development".

7. In line 68, should be written with italic - Fusarium oxysporum.

8. In line 93 it should be noted that IbSWEET17 is localized on the outer (or inner ?) membrane of the chloroplast. Or at least: on chloroplast membrane.

9. In line 94, is correct MtN3/saliva (PQ-loop repeat) ...?

10. In line 113 (the name of Figure 2): replace "lacation" with location.

11. In line 145, when you first mention XGH and GJ8, you specify that these are two cultivars of sweet potato abbreviated with XGH and GJ8, even if you mention this in "Material and Method".

12. In line 155, I recommend that the name of Figure 5 be: Sugar, starch content and weight of storage root changes in two sweet potato cultivars in different periods.

13. In line 179, Figure 6, name should be: IbSWEETs expression analysis of different sites at two sweet potato cultivars in different periods.

14. In lines 218-222: The title of Table 3 ends with GJ8. The following lines refer to the stress induced by NaCl treatment. Therefore, move these lines after paragraph that refers to the treatment with 4 oC.

15. In line 226, correct the with The (beginning of the phrase) relative expression ...

16. In line 361, it is correct: The sweet potato - Ipomoea batatas (L.) Lam cultivars ...

17. In line 381, in the given context, Arabidopsis thaliana should be written in italic.

18. In line 397, should: Chromatographic conditions: Dionex CarboPac PA1...

I wish you success in publishing this study!

Author Response

Dear Reviewer:

Thank you very much for taking the time to review this manuscript entitled “Molecular cloning, expression analysis and functional analysis of 9 IbSWEETs in Ipomoea batatas (L.) Lam” (ID: ijms-2674942). Those comments are all valuable and very helpful for revising and improving our paper, as well as the important guiding significance to our researches. We have studied comments carefully and have made correction which we hope meet with approval. Revised portion are marked in red in the paper.

2. Questions for General Evaluation

Reviewer’s Evaluation

Response and Revisions

Does the introduction provide sufficient background and include all relevant references?

Yes

Thank you

Are all the cited references relevant to the research?

Yes

Thank you

Is the research design appropriate?

Yes

Thank you

Are the methods adequately described?

Yes

Thank you

Are the results clearly presented?

Yes

Thank you

Are the conclusions supported by the results?

Yes

Thank you

3. Point-by-point response to Comments and Suggestions for Authors

Comments 1: In the title of the manuscript, the scientific name should be complete, ie: Ipomoea batatas (L.) Lam

Response 1: Thank you. We have corrected it. This change can be found in title and in line 386 in the revised manuscript.

Comments 2: Given the multitude of data, the Abstract should be more consistent.

Response 2: Thank you. We have re-written this section. This change can be found in lines 17-29 in the revised manuscript.

Comments 3: The objectives and the aim of the research are not clearly outlined.

Response 3: Thank you. We have added this content. The objectives and the aim of the research was “To gain a more comprehensive and deeper understanding of the role of SWEET genes in the growth, development and sugar accumulation of sweet potato”. This change can be found in lines 86-88 in the revised manuscript.

Comments 4: In line 27, you probably wanted the phrase to start with "Sucrose, as the main ..". Otherwise, the phrase would be too long!

Response 4: Thank you. We have revised the sentence as “Sucrose is the main carbohydrate delivering from photosynthetic tissues to heterotrophic tissues, and it is also the core of the resource allocation system.” This change can be found in lines 35-37 in the revised manuscript.

Comments 5: In line 50, Mainly should be mainly. The next sentence to begin with: Loss of AtSWEET13 and ...

Response 5: Thank you. We have corrected it. This revise can be found in lines 58-59 in the revised manuscript.

Comments 6: In lines 54-55, you omited the verb: "In addition, SISWEET5b [25],....and so on are also associated with flower development".

Response 6: Thank you. The verb "are" was added to this sentence. This change can be found in line 63 in the revised manuscript.

Comments 7: In line 68, should be written with italic - Fusarium oxysporum.

Response 7: Thank you. We have corrected it. This revise can be found in line 84 in the revised manuscript.

Comments 8: In line 93 it should be noted that IbSWEET17 is localized on the outer (or inner ?) membrane of the chloroplast. Or at least: on chloroplast membrane.

Response 8: Thank you. IbSWEET17 is localized on the membrane of the vacuoles, not of the chloroplast.

Comments 9:  In line 94, is correct MtN3/saliva (PQ-loop repeat) ...?

Response 9: Thank you. We have revised MtN3/saliva (PQ-loop repeat) as “MtN3 / saliva domain (CDD accession No. pfam03083) or PQ - loop superfamily (CDD accession No. cl21610)”. This revise can be found in lines 113-114 in the revised manuscript.

Comments 10: In line 113 (the name of Figure 2): replace "lacation" with location.

Response 10: Thank you. It has been corrected. This revise can be found in line 133 in the revised manuscript.

Comments 11: In line 145, when you first mention XGH and GJ8, you specify that these are two cultivars of sweet potato abbreviated with XGH and GJ8, even if you mention this in "Material and Method".

Response 11: Thank you. We have revised the “XGH and GJ8” as “the sweet potato cultivars “xiguahong” (XGH) and “guijingshu 8” (GJ8)”. This change can be found in lines 165 in the revised manuscript.

Comments 12: In line 155, I recommend that the name of Figure 5 be: Sugar, starch content and weight of storage root changes in two sweet potato cultivars in different periods.

Response 12: Thank you. We have revised the name of Figure 5 according to your suggestion. This change can be found in lines 176-177 in the revised manuscript.

Comments 13: In line 179, Figure 6, name should be: IbSWEETs expression analysis of different sites at two sweet potato cultivars in different periods.

Response 13: Thank you. We have revised the name of Figure 6 according to your suggestion. This change can be found in line 200 in the revised manuscript.

Comments 14:  In lines 218-222: The title of Table 3 ends with GJ8. The following lines refer to the stress induced by NaCl treatment. Therefore, move these lines after paragraph that refers to the treatment with 4 oC.

Response 14: Thank you. It has been revised according to your suggestion. This change can be found in lines 241-244 in the revised manuscript.

Comments 15: In line 226, correct the with The (beginning of the phrase) relative expression ...

Response 15: Thank you. We have corrected it. This change can be found in line 248 in the revised manuscript.

Comments 16:  In line 361, it is correct: The sweet potato - Ipomoea batatas (L.) Lam cultivars ...

Response 16: Thank you. We have corrected it. This change can be found in line 386 in the revised manuscript.

Comments 17: In line 381, in the given context, Arabidopsis thaliana should be written in italic.

Response 17: Thank you. We have corrected it. This change can be found in line 408 in the revised manuscript.

Comments 18:  In line 397, should: Chromatographic conditions: Dionex CarboPac PA1...

Response 18: Thank you. We have revised it as “Chromatographic conditions”. This change can be found in line 428-429 in the revised manuscript.

Special thanks to you for your good comments.

Round 2

Reviewer 2 Report

Comments and Suggestions for Authors

The authors have suitably incorporated all my suggestions in the revised manuscript. Now the manuscript is technically looks good. However, the authors need to consider the following comments to improve the quality of this manuscript.

Line 35-37: Syntax error. Reframe the lines.

Line 86: should be ‘Considering the important role of SWEETs’

Figure 1 is not clear enhance it.

In figure 7, write either “relative fold change in gene expression or Relative gene expression”. This point should be there in figure not in figure legend. The previous figure legend is fine.

Insert the ExPASy Protparam database reference in section 4.2.

Line 422: Do not start the line with numbers.

Line 460: If you are using correlation the next linker will be ‘between’. Check it.

Line 460: Should be ‘sweet potatoes’

Comments on the Quality of English Language

Revised lines seems syntax errors.  Please improve the same with the points mentioned. 

Author Response

Thank you so much for making another comment on our manuscript entitled “Molecular cloning, expression analysis and functional analysis of 9 IbSWEETs in Ipomoea batatas (L.) Lam” (ID: ijms-2674942). Those comments are all valuable and very helpful for revising and improving our paper. We have studied comments carefully and have made correction which we hope meet with approval. Revised portion are marked in blue in the paper.

2. Questions for General Evaluation

Reviewer’s Evaluation

Response and Revisions

Does the introduction provide sufficient background and include all relevant references?

Yes

Thank you

Are all the cited references relevant to the research?

Yes

Thank you

Is the research design appropriate?

Yes

Thank you

Are the methods adequately described?

Yes

Thank you

Are the results clearly presented?

Yes

Thank you

Are the conclusions supported by the results?

Yes

Thank you

3. Point-by-point response to Comments and Suggestions for Authors

Comments 1: Line 35-37: Syntax error. Reframe the lines.

Response 1: Thank you. We have re-written this section. This revise can be found in lines 35-36 in the revised manuscript.

Comments 2: Line 86: should be ‘Considering the important role of SWEETs’

Response 2: Thank you. We have modified it. This revise can be found in line 85 in the revised manuscript.

Comments 3: Figure 1 is not clear enhance it.

Response 3: Thank you. We have redrawn and replaced it. This revise can be found in line 117 in the revised manuscript.

Comments 4: In figure 7, write either “relative fold change in gene expression or Relative gene expression”. This point should be there in figure not in figure legend. The previous figure legend is fine.

Response 4: Thank you. We have modified it. This revise can be found in lines 249-254 in the revised manuscript.

Comments 5: Insert the ExPASy Protparam database reference in section 4.2.

Response 5: Thank you. We have cited the corresponding references for ExPASy Protparam database. This revise can be found in line 403 in the revised manuscript.

Comments 6: Line 422: Do not start the line with numbers.

Response 6: Thank you. We have modified it. This revise can be found in lines 420-421 in the revised manuscript.

Comments 7: Line 460: If you are using correlation the next linker will be ‘between’. Check it.

Response 7: Thank you. We have modified it. This revise can be found in line 458 in the revised manuscript.

Comments 8: Line 460: Should be ‘sweet potatoes’

Response 8: Thank you. We have modified it. This revise can be found in line 458 in the revised manuscript.

Once again, thank you very much for your comments.
